# Pigs and pasture: Drivers and characteristics of outdoor systems on the island of Ireland

Ophelie Menant[1]*, Laura Boyle[1], Siobhan Mullan[2], Fidelma Butler[3], Keelin O'Driscoll[1]

**1** Pig Development Department, Animal and Grassland Research and Innovation Centre, Teagasc Moorepark, Fermoy, Co Cork, Ireland, **2** University College Dublin, Dublin, Ireland, **3** School of Biological, Earth and Environmental Sciences, University College Cork, Cork, Ireland

\* ophelie.menant@gmail.com

## Abstract

Outdoor systems offer benefits to pig welfare but they also pose challenges and are less well researched than indoor. This study characterized, for the first time, current husbandry and management practices of outdoor pig production on the island of Ireland, in order to understand associated drivers, challenges and changes needed for sustainable development. An online survey conducted from December 2022 to April 2023 (n = 90 respondents) revealed that animal welfare, food quality, and traceability concerns were the primary reasons for raising pigs outdoors, regardless of whether the pigs were raised for sale (meat or live pigs) or other (e.g., personal meat consumption, pet, land management) purposes. Most of the respondents expressed concerns about animal welfare in conventional systems, and emphasized the importance of the "Five Freedoms". A significant cohort of respondents (56%) adopted alternative health management strategies based on season and soil management (paddock rotation, use of straw bedding on muddy areas), and by avoiding the use of antibiotics and other chemicals. Farms were small (median: 20.5 pigs, 1.02 ha), half were engaged with pig societies, and used traditional/rare breeds, which are adapted to outdoor conditions and that farmers can sell as superior quality pigmeat. The most frequently reported challenges were feed costs, inclement weather, fencing and soil maintenance. The most mentioned "needs" of the industry were financial support and consumer education. While participants chose to raise pigs outdoors to improve animal welfare, meat quality and traceability; overcoming challenges related to finance, infrastructure, and education is vital to their future sustainability.

## Introduction

In Ireland, pigmeat production is the third largest agricultural sector [1] with more than 50% of pig meat exported worldwide [2]. Similar to the rest of the EU, Irish citizens express concerns about animal welfare [3]. According to the 2023 Irish National Pig Census, there were 1372 active pig herds in the Republic of Ireland, of which, 75.2%

**Data availability statement:** Raw data and its associated code book is available at https://opendata.teagasc.ie/dataset/survey_production_health_welfare_outdoor_pig_farms_ireland.

**Funding:** This study was funded by the Department of Agriculture, Food and the Marine in Ireland (Grant number 2021R600). The funders had no role in study design, data collection and analysis, decision to publish, or preparation of the manuscript.

**Competing interests:** The authors have declared that no competing interests exist.

kept less than 100 pigs; with 65.2% of herds less than 5 pigs [4]. Similarly, in Northern Ireland in 2023, 54.7% of 369 farms kept less than 100 pigs, with 39% less than 9 pigs [5]. These large number of very small pig herds may include outdoor systems, yet very little is known about them.

In the EU 25% of pigs are raised in non-intensive systems and generally raised outdoors in more traditional systems categorised as backyard or extensive [6]. In countries such as Romania and Hungary, 99% of pig farms follow a backyard model, where pigs are raised for personal meat consumption and for the local community, with an average number of 3.3 to 21.4 pigs per farm [7]. In many other European countries, the proportion of pigs produced outdoors is < 5% [8] but in Spain about 5% of production is of local breeds raised extensively as a component of a high value meat market [6,9]. While outdoor systems can be used to raise pigs for personal meat consumption and commercial purposes, pigs are also kept outdoors in rescue centres and/or in educational farms [10], or as pets. The reasons for keeping pigs outdoors are diverse, as are characteristics of how they are managed, such as the outdoor underfoot surface (forest vs pasture vs concrete), size of the outdoor area available for pigs, management practices, breeds used, and their organic status [11].

Outdoor pig production systems are associated with several animal welfare advantages over intensive indoor systems, primarily because pigs have more space and access to a natural environment allowing them greater behavioural freedom [12]. Furthermore, there is generally a reduced risk of physical discomfort and injury and reduced need for tail docking and teeth clipping in outdoor systems [13]. These systems are often associated with lower capital investments and have been reported to offer societal benefits that bring satisfaction to farmers [14]. However, outdoor systems face challenges, such as ensuring that pigs have consistent access to food and water, managing their exposure to inclement weather, and mitigating biosecurity risks from wildlife [13,14]. Outdoor systems may produce higher greenhouse gas emissions than conventional systems (Gt or kg $CO_2$ equivalent, [15,16]), and soil health can be impacted depending on paddock management and animal density [17]. Additionally, as these systems are often small-scale they may not qualify for financial support schemes [14].

Although intensive and organic pig farm systems are well described and studied, small and/or outdoor ones are not. In order to address this knowledge gap about outdoor pigs raised on the island of Ireland, we developed an online questionnaire for pig owners which included questions on farm structure and animal management. We also aimed to understand pig owners' rationale for entering small scale pig production, their perceptions of pig health and welfare, and of the challenges and changes needed to support them in raising pigs outdoors.

## Materials and methods

This survey was approved by the Social Research Ethics committee of University College Cork (Ethical approval No. 2022−217). Participants were informed in writing that their participation was voluntary, that their information would be treated confidentially and pseudo-anonymously, and that it would be used solely for the development

of the OneWelPig project. Consent was obtained by asking participants to confirm these statements and affirm that they were over 18 years of age by ticking a box at the beginning of the survey. Participants could not progress to the survey questionnaire unless the box was ticked.

## Survey design

The online survey for outdoor pig herd owners/managers was developed for distribution within the online platform SurveyMonkey (https://www.surveymonkey.com/). Forty questions covered the following topics (See S1 Text for full questionnaire):

- Demographics and general farm information (11 questions) which include a multiple-choice question about whether respondents raised pig to an organic or GM free standard, but not if they had a certification.

- Husbandry (8 questions),

- Feed and water management (8 questions),

- Animal health management (5 questions) which included two distinct questions about the pigs' health conditions, diseases and parasites. Health conditions were defined as physical or visual symptoms that farmers could observe on their pigs, while diseases and parasites were a list of viruses, bacteria and parasites known to induce health issues in pigs.

- Farmer attitudes and perceptions of the outdoor pig industry (8 questions) where respondents were asked to rank the following 7 attributes from "not at all important" to "extremely important": Animal welfare, societal concerns, market opportunity, reduced costs, environmental sustainability and management, animal food provenance, and meat quality. This section then concluded with the following open questions where respondents were asked to use one individual word or expression per answer:

  1. What are the three main daily challenges of keeping pigs outdoors on your farm?

  2. What are three main changes that are needed at infrastructural level to encourage the production of pigs outdoors?

  3. How would you describe your farm?

  4. What do your pigs really enjoy in life?

  5. What do your pigs least enjoy in life?

## Survey distribution

The link to the survey was circulated via public posts on social networks (X/Twitter, Instagram, Facebook), the Teagasc webpage and newsletter, via personal messages that were sent to potential participants identified on social networks, via company emails and phone numbers that were freely available online, the Irish Pig Society, and agricultural shows. Furthermore, snowball sampling was used by encouraging participants to share the survey to colleagues. Data were collected via the online platform from 26 December 2022–9 April 2023. Responses were considered valid when submitted by persons managing pigs outdoors on the island of Ireland.

## Data management

A total of 123 responses were collected, of which 90 were considered suitable for analysis. These were classified as responses from producers that raised pigs for sale purpose (produce pigmeat and/or live pigs for sale, n = 49) and owners of pigs for purposes other than sale (personal meat consumption, land management, breed conservation, hobby, tourism

and showing pigs, n = 41). Participants that did not answer the 8 eligibility questions (questions annotated with a * in the S1 Text) or who submitted multiple replies were not included. For the open questions, answers that were considered irrelevant (e.g., "none", "daft question", "ditto" …) were removed. When a participant gave the same exact answer 3 times to the same question, only one was retained (e.g., "cold, cold, cold"). Data management was performed using Excel ® (Micrososft ®, 2016). Overall data can be found in the Teagasc repository centre (https://opendata.teagasc.ie/dataset/survey_production_health_welfare_outdoor_pig_farms_ireland), and detailed responses are provided in S1 Text.

### Open questions analyses

KOD first read the answers from the open questions, assigned them codes, and subcategorised some codes with explainers. Codes were words or phrases to cluster responses into related categories and thus allowed the data to be summarised. They were identified from reading and re-reading the replies to the open questions. Answers that contained synonyms of the same words (e.g., movement, moving pigs) were given the same code (e.g., animal management). Codes could also include words that are not synonyms, but refer to similar topics (e.g., 'a good fence' and 'breakouts' were both assigned the code 'fencing'). Explainers were additional words that provided more details than the code (e.g., 'Wet' and 'Hot' were explainers for the code 'Weather'). More than one code could be assigned to an answer (e.g., 'Fresh food and water' was assigned to both of the codes 'Food' and 'Water'), and multiple explainers could be assigned to one code. Codes and explainer labels were then reviewed and refined by OM and LB. Discrepancies were discussed and resolved by KOD and OM. The final number of codes for each question was then counted, and their frequency of occurrence (%) calculated by dividing the number of occurrences of each code by the number of responses, then multiplying by 100. The frequency of occurrence of explainers was calculated by dividing the number of occurrences of each explainer by the total number of times its related code occurred, then multiplied by 100.

### Data analysis

All statistical analyses were performed using RStudio (2022.07.02 version). Co-occurrence tests were performed on two multiple choice questions: 1) the purpose of raising pigs outdoors, and 2) the nature of the underfoot surface, using the *co-occur* package [18]. The *visNetwork* package [19] was used to visually represent the results. To analyse the 5 open questions, co-occurrence tests were also performed on the count of the codes identified for each answer. Only results when observation occurrences were higher than expected are reported. Due to the diversity in farm systems, no farm characteristics were included as confounding variables in the co-occurrence analyses. The rating question was analysed using the *likert* function from the "H.H" package [20]. To compare the responses between owners raising pigs for sale and other purposes, Fisher tests were carried out on categorical questions, and the Fisher exact test was used as a post hoc test to compare the proportion of answers within each answers' option. Wilcoxon tests were performed on numerical open questions. All p values were two-tailed, and a p value < 0.05 was considered statistically significant.

## Results

### Demographics and general farm information

Responses to the survey came from 24 of the 32 counties in Ireland (22 from the Republic of Ireland and 2 from Northern Ireland). Respondents stated they mostly raised outdoor pigs for personal meat consumption (59/90), to produce meat for sale (49/90) followed by producing pigs for sale (35/90) and, to a lesser extent, for land management purposes (30/90) and breed conservation (29/90). Pigs were also kept as pets or as a hobby (23/90), for tourism (9/90), for livestock shows (9/90), having been rescued (2/90) and finally "other" purposes (4/90, answers not mutually exclusive) (Fig 1). Personal meat consumption co-occurred more than expected with keeping pigs for land management (26 observed vs 19.7 expected co-occurrences, p = 0.002) and breed conservation (24 observed vs 19 expected co-occurrences, P < 0.05). The purpose

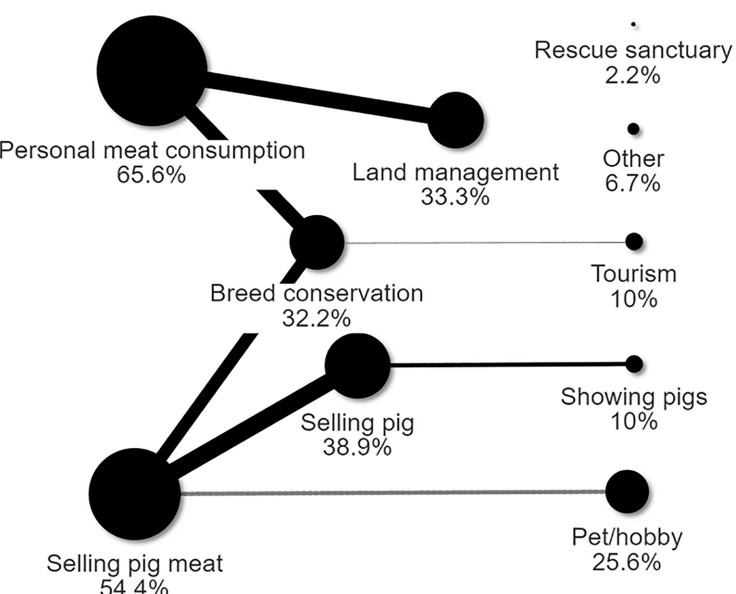

**Fig 1. The stated purpose for raising pigs outdoors (% farms, n=90).** Black lines represent significantly higher than expected co-occurrences. Grey lines represent significantly lower than expected co-occurrences. The thickness of the line represents the number of co-occurrences.

of selling meat co-occurred more than expected with breed conservation (20 observed vs 15.8 expected co-occurrences, p=0.046) and selling pigs (28 observed vs 19.1 expected co-occurrences, P<0.001), this latter co-occurring also more than expected with showing pigs (7 observed vs 3.5 expected co-occurrences, p=0.016). The purpose of breed conservation co-occurred less than expected with tourism (0 observed vs 2.9 expected co-occurrences, p=0.025), and selling pig meat co-occurred less than expected with pet or hobby purposes (7 observed vs 12.5 expected co-occurrences, p=0.007).

Of 44 out of the 49 participants who raised pigs for the purpose of selling meat, 81.8% sold it directly from the farm, 22.7% at a farmer market, 20.5% at a butcher shop, 20.5% online shop, 4.5% at supermarket chain and 18.2% at another location (answers not mutually exclusive). Of the 49 participants who raised pigs outdoors to sell the meat, 83.7% slaughtered their pigs at a licenced slaughterhouse, 20.4% at a butcher/victualler and 4.1% at other places (answers not mutually exclusive). The selection of slaughter facility was based upon proximity to the farm (54.4%), and whether they facilitated slaughter of small numbers of pigs (45.6%). Other, less frequently stated reasons included having good animal welfare standards and organic certification (8.7% for both answers), being the only place that accepted their pigs (15.2%) or for offering the cheapest slaughter rate (6.5%, answers not mutually exclusive).

Of the 74 respondents who answered the question, 93.2% were registered with a department of agriculture (i.e., they had a herd or flock number), whereas 4.05% preferred not to say and 2.7% were not registered. There was a significant association between the purpose of keeping pigs and being registered (p=0.019). The number of participants with a herd number was higher when pigs were kept for sale purpose (59.4%) than for other purposes (40.6%, p=0.04, odd ratio=2.1).

Of all 90 respondents, 38.9% of the participants reported they raised pigs to an organic standard, 11.1% to a GM free standard and 50% did not raise pigs to any standard. There was a significant association between the purpose of raising pigs and complying with a particular standard (p=0.03). The number of participants raising pigs to an organic or GM free standard was higher when pigs were kept for sale purposes (66.6%) than for other purposes (33.3%, p=0.003, odds ratio=0.259).

Across the 90 responses, 55.6% of farms had only 1 person in charge of the pigs. The remaining 44.4% of farms had 3.2±2.7 persons in charge of the pigs. Half of the participants had approx. 7 years of experience in producing pigs

outdoors. There was no difference between farms that kept pigs for sale or another purpose on the number in charge of the pigs nor the number of years in the industry.

Of the 90 respondents, 52.2% of the participants were members of at least one pig society, 44.4% were not and 3.3% preferred not to say. There was an association between the purpose for which participants kept pigs, and whether they were members of a society (p=0.002); more participants who kept pigs for sale were part of a society (62.9%) than those who kept pigs for another purpose (37.1%, p<0.001, odd ratio=0.18).

## Husbandry

The median of the highest number of pigs kept at any one time was 21 (range: 1–277). This median herd size was higher on farms where pigs were kept for the purpose of sale (median: 29.5, range: 2–277) than for other purposes (median: 6, range: 1–111, z=−4.89, p<0.0001). The number of boars and sows/gilts was similar, whether kept for sale or other purposes (Fig 2). However, farms that kept pigs for sale had significantly more growers and unweaned pigs than farms where pigs were kept for other purposes (Fig 2).

Of 89 responses, 57.3% of the participants bred pigs on their own farm, 47.2% sourced pigs from other outdoor farms, and a smaller proportion from internet or newspaper adverts (22.6%) and indoor farms (12.4%; answers not mutually exclusive). There was an association between the purpose of keeping pigs and the source from which they came (P<0.001). More participants bred pigs on their own farm if the purpose was for sale (78.4%) than for other purposes (21.6%, p<0.001, odd ratio=12.8).

Across the 90 responses, a total of 22 different pure breeds were identified. The main breeds kept were Duroc and Oxford Sandy and Black (OSB) (33.3% each), followed by Tamworth (24.4%), Gloucester old spot (GOS) (20%), Kune-Kune (15.6%), then Landrace (14.4%), British Saddleback, and Berkshire (13.3% each, answers not mutually exclusive).

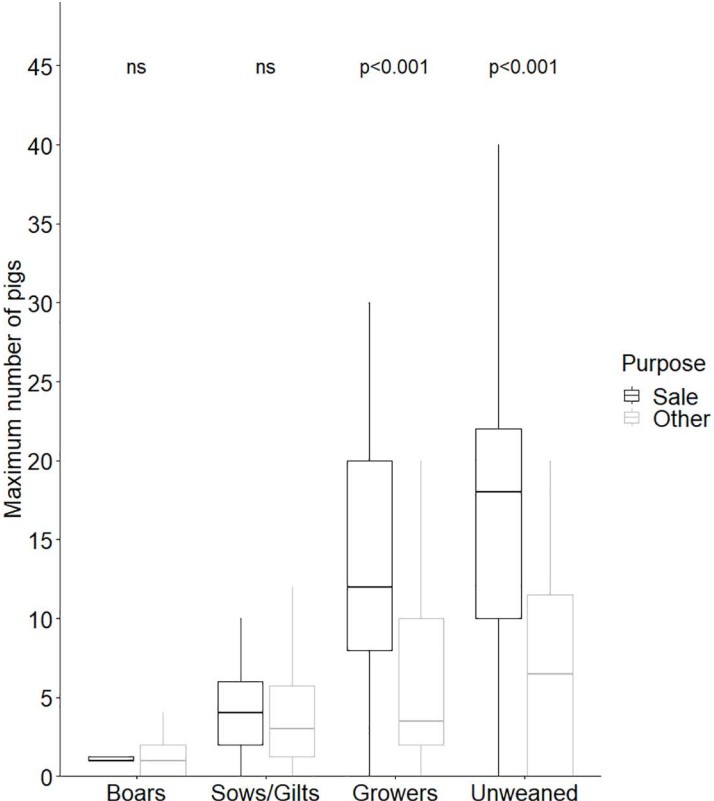

**Fig 2. Median of the highest number of pigs raised outdoors at any one time, for sale (black bars) or other purposes (white bars).** (n=88 respondents).

There was an association between the purpose of keeping pigs and the breed (P<0.01) and between the purpose and the sex (p=0.006). When participants raised pigs for sale purpose, there was a significant association between the breed and the sex of the pigs (p=0.036). The number of participants that raised Duroc was higher for boars (75%, n=12) than sows/gilts (25%, n=4, p=0.012). The number of participants that raised OSB, GOS and Tamworth was lower for boars (respectively 14.3% n=3, 20% n=2, 20% n=2) than sows/gilts (respectively 85.7% n=18, p<0.001, 80%, n=8 p=0.023, 80% n=8, p=0.023).

Based on 84 responses, the median area used for keeping pigs was 10,235 m² (range: 16–3 237 000 m²). This was greater when pigs were kept for sale (median: 16187, range: 140–3 237 000 m², n=46) rather than other purposes (median: 6071, range: 16–80937 m², z=−4.440, p<0.0001, n=38). Of 87 respondents, the soil type mainly reported was clay (77%), followed by loam (33.3%), sand (10.4%) and silt (9.2%, answers not mutually exclusive), with no association with the purpose of keeping pigs. The underfoot surfaces were mainly paddocks or fields with vegetation (72/88), scrubland (33/88), and woodland (29/88), followed by additional straw (18/88), mucky area (13/88), solid concrete (8/88), tillage soil (6/88) and concrete slats and other (2/88 for both, answers not mutually exclusive), again with no association with the purpose of keeping pigs (Fig 3). Areas of scrubland co-occurred more than expected with woodland (15 observed vs 10.6 expected co-occurrences, p=0.036). The use of additional straw co-occurred more than expected with mucky areas (6 observed vs 2.6 expected co-occurrences, p=0.02) and solid concrete (6 observed vs 1.6 expected co-occurrences, p<0.001).

Of 87 respondents, all the participants reported that pigs had access to shelter, mainly mobile huts (60.9%), followed by buildings that were continuously open (40.2%) and natural shelter (37.9%), then fixed huts (25.35%), and buildings opened at specific hours (11.5%).

## Feed and water management

Of 88 respondents, participants stated that pigs were mainly fed outdoors using several moveable feeders (48.8%) or scatter fed (38.4%). Feeders were reported at lower proportions outdoors in a fixed (18.6%) or single location (14%), than

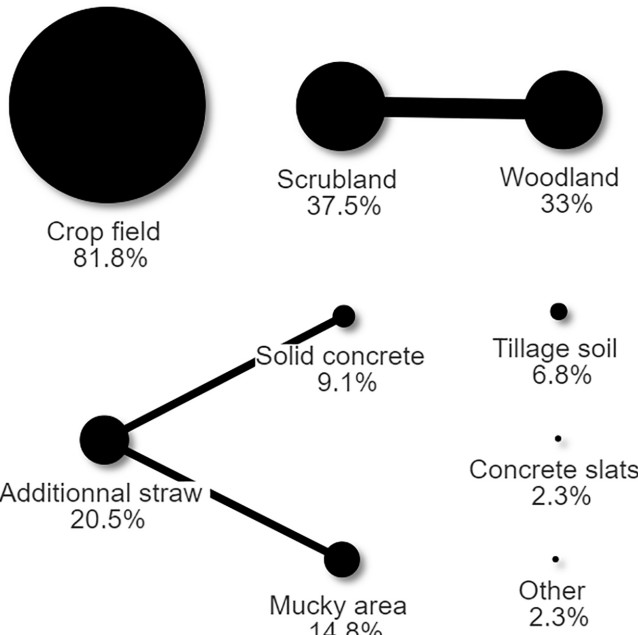

**Fig 3. Proportion of farms that raised pigs outdoors depending on the underfoot surface.** Black lines represent when observed co-occurrences' were significantly higher the expected co-occurrences. The thickness of the line represents the number of co-occurrence.

indoors located in several areas (10.5%) or single area (3.5%, answers not mutually exclusive). There was an association between the feeding location and the purpose of keeping pigs (p = 0.036). Of 85 respondents, 63.5% stated that feeding strategies were based on the season. Of 86 respondents, participants reported that feed was mainly sourced from merchants (88.4%), grown at home (22.1%) and from other sources (10.5%, answers not mutually exclusive). There was no significant association between the purpose of keeping pigs outdoors and the feeding strategies, nor the feed source.

Of 84 respondents, participants reported that drinkers were mainly located outdoors in a single place (38.1%), in several fixed places (35.7%) or several movable places (26.2%), with indoor drinkers less frequently used (located in several places (21.4%) or in a single location (14.3%), answers not mutually exclusive). There was an association between the drinker location and the purpose of keeping pigs (p = 0.03).

### Animal health management

Of 83 respondents, just over two thirds (68.7%) reported having no health conditions on their farms, or concerns about mortality rates (0% for adult pigs and piglets). A low proportion of respondents reported heat stress (10.8%), lameness (9.6%), overweight pigs (4.8%), skin lesions (4.8%), reproduction difficulties (4.8%), cold stress (3.6%), gastrointestinal disorders (3.6%), respiratory symptoms (3.6%), poor body condition (2.4%), dehydration (1.2%), urinary issues (1.2%), and watery eyes (1.2%, answers not mutually exclusive). There was no association between the reported health condition of the animals and the purpose for which they were kept.

Regarding disease and parasites, of 42 participants, 31% of respondents reported none, 23.8% did not know, 23.8% reported skin parasites, and 21.4% gastrointestinal parasites (answers not mutually exclusive). Only 2.4% to 7.1% participants reported diseases other than parasites on their farms. There was no association between the disease and parasites observed in the previous 12 months on the farms and the animals age (unweaned vs weaned), or the purpose of raising pigs.

To prevent and treat diseases, parasites and/or lesions, of 76 respondents, participants mainly reported using good animal care (93.4%), good hygiene and equipment maintenance (64.5%), and performed paddock rotation (63.2%). Lesser used methods included anthelmintics (25.0%), antibiotics (19.7%), quarantine (17.1%), vaccines (14.5%), plants (10.5%), homeopathy (10.5%), pain relief (7.9%) and antiseptics (6.6%, answers not mutually exclusive). There was no association between the practices to prevent and treat the animals, and the purpose of keeping pigs.

Of 85 respondents, participants mainly sought advice from one veterinarian (51.8%) followed by other outdoor farmers (29.4%); a lower proportion used several veterinarians (8.2%), or other sources of advice (9.4%, including societies). None reported seeking advice from farmers producing pigs indoors. There was an association between the source of advice and the purpose of raising pigs outdoors (p = 0.009). Of participants who asked for advice from a single veterinarian, more did so when pigs were kept for sale (61.4%) than for other purposes (38.6%, p = 0.05, odds-ratio = 2.5). Of the 51 respondents, participants mainly consulted general farm/large animal specialists (56.9%), then mixed (small and large) specialists (25.5%) and specialists with strong experience/knowledge of outdoor pig production (21.6%). A lower proportion consulted specialists with strong experience/knowledge of conventional pig production (7.8%) and small animal specialists (2%).

### Farmer attitudes and perceptions of the outdoor pig industry

**I produce pigs this way because….** Details of the attributes that respondents considered important in relation to keeping pigs outdoors are shown in Fig 4. The six reasons that were considered "extremely important" were: "I believe it offers better animal welfare" (91.8%), "I want my pigs to access the 5 freedoms of animal welfare" (89.3%), "it is safer, tastier and better quality meat" (81.4%) and "I have concerns about conventional pig production" (68%), "food provenance is important for me" (66.7%), and "I believe it is more environmentally sustainable" (58.7%).

The proportion of participants that considered "seeing an opportunity in the market for pig meat" depended on the purpose (p < 0.0001). The proportion that answered "extremely important" was greater amongst those who kept pigs for sale

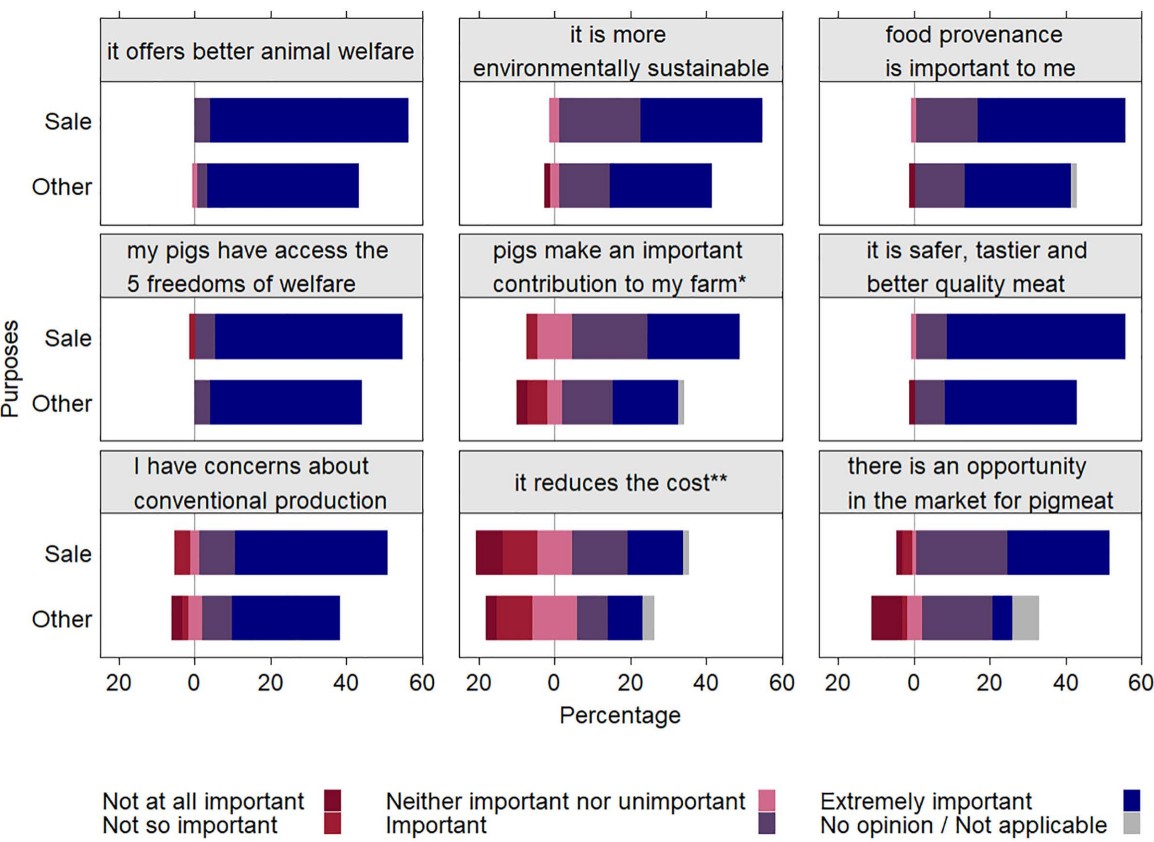

**Fig 4. Percentage of rating answers to the question "I produce pigs outdoors because…" depending on the purpose (sale vs other, n=75 responses in total).** *: soil health, they eat organic waste etc, **: cost of the feed and building's maintenance.

(83.3%) than for other purposes (16.7%, p<0.0001, odds-ratio=22.6). Participants that answered "Not at all important" for this reason were fewer when pigs were kept for sale (14.3%) than for other purposes (85.7%, p=0.03, odds-ratio=0.04).

**What are the main daily challenges of keeping pigs outdoors on your farm?** Of 66 respondents, 20 codes were identified and assigned to 174 responses. The 7 most used codes were related to feed (n=35), fencing (n=33), weather (n=31), costs (n=20), labour and time (n=16), paddock management (n=16), and animal management (n=15, answers not mutually exclusive, Fig 5A). The "Feed" code significantly co-occurred with the "Costs" code (17 observed vs 4 expected co-occurrences, p<0.0001), and the "Fencing" code with the "Maintenance" code (6 observed vs 1.3 expected co-occurrences, p<0.0001, maintenance: n=7 or 4%, Fig 5A). The "Weather" code was mainly associated with the explainers wet/rain (n=5), followed by dry, cold and sun conditions (n=3 each), and finally winter (n=2, answers not mutually exclusive, Fig 5B). Paddock management was related to the paddock rotation explainer (n=7, Fig 5B).

**What are the changes needed at infrastructural level?** A total of 60 respondents answered this question, and 15 codes were assigned to 151 responses. The 6 main codes were related to education (n=54), knowledge sharing (n=25), financial support (n=19), marketing (n=12), regulatory body (n=12), and policy (n=11, answers not mutually exclusive, Fig 6A). There was no significant co-occurrence between the codes related to the changes needed. Participants reported a need for education of both consumers (n=13), and producers (n=4), mostly in relation to animal welfare (n=13) and the environment (n=3, answers not mutually exclusive, Fig 6B). Participants reported a need of knowledge sharing via advisory services (n=8), peer-to-peer learning (n=7), and via veterinarians (n=1, answers not mutually exclusive,

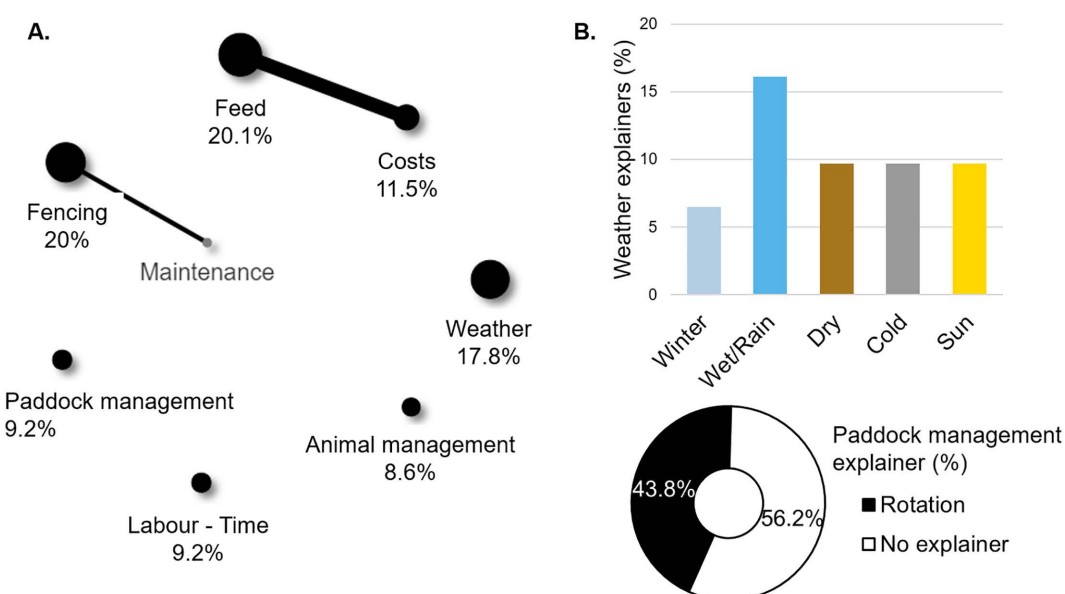

**Fig 5. Proportion of the 7 codes mostly used to describe the main challenges of raising pigs outdoors (black dots, A.) and their associated explainers (B.).** Black lines represent when observed co-occurrences were significantly higher the expected co-occurrences. The thickness of the line represents the number of co-occurrences. Grey dots represent a code that highly co-occurred with one of the mostly used codes without being mostly used itself.

Fig 6B). Finally, participants reported a need for financial support for infrastructure (n = 4) and feed (n = 1, answers not mutually exclusive, Fig 6B)

**How would you describe your farm?** Of 66 respondents, 50 codes were assigned to 170 responses. The 10 main codes used to describe the farm were enjoyable (n = 17, 10%), small (n = 16, 9.4%), high welfare (n = 13, 7.6%), mixed (n = 12, 7.1%), biodiversity (n = 10, 5.9%), non-intensive (n = 9, 5.3%), organic (n = 8, 4.7%), sustainable (n = 7, 4.1%), smallholding (n = 7, 4.1%), and traditional (n = 6, 3.5%, answers not mutually exclusive). There was no significant co-occurrence between the codes related to the farm description.

**What do your pigs really enjoy in life?** Of 67 respondents, 26 codes were assigned to 187 responses. The 6 main codes related to freedom (n = 35), food (n = 29), rooting (n = 22), company (n = 21), wallowing (n = 14), and weather (n = 11, answers not mutually exclusive, Fig 7A). There was no significant co-occurrence between the different codes related to what pigs like. The "Freedom" code was associated with the explainers roaming behaviour (n = 13) and social group (n = 2, answers not mutually exclusive, Fig 7B). The "Food" code was associated mostly with fruits and veggies (n = 6), and food quantity (n = 2, answers not mutually exclusive, Fig 7B). Participants reported that pigs enjoy the company of humans (n = 9) and pigs (n = 7, answers not mutually exclusive, Fig 7B). The "Weather" code was mainly related to sunny conditions (n = 9), and summer (n = 1, answers not mutually exclusive, Fig 7B).

**What do your pigs least enjoy in life?** Of 61 respondents, 142 responses were assigned to 23 codes. The 5 main codes were "Weather" (n = 44), "Nutrition issues" (n = 16), "Confinement" (n = 15), "Moving" (n = 10), and "Change" (n = 10, answers not mutually exclusive, Fig 8A). There was no significant co-occurrence between the codes related to what pigs dislike. "Weather" was mainly associated with the explainers wet/rain (n = 16), followed by cold (n = 14) and hot (n = 11), and at a lesser extent with wind (n = 2) and winter (n = 1, answers not mutually exclusive, Fig 8B). The nutrition issues reported by the participants were related to the explainers hunger (n = 10) and thirst (n = 2, answers not mutually exclusive, Fig 8B). Participants stated that pigs dislike social changes (n = 6, Fig 8B).

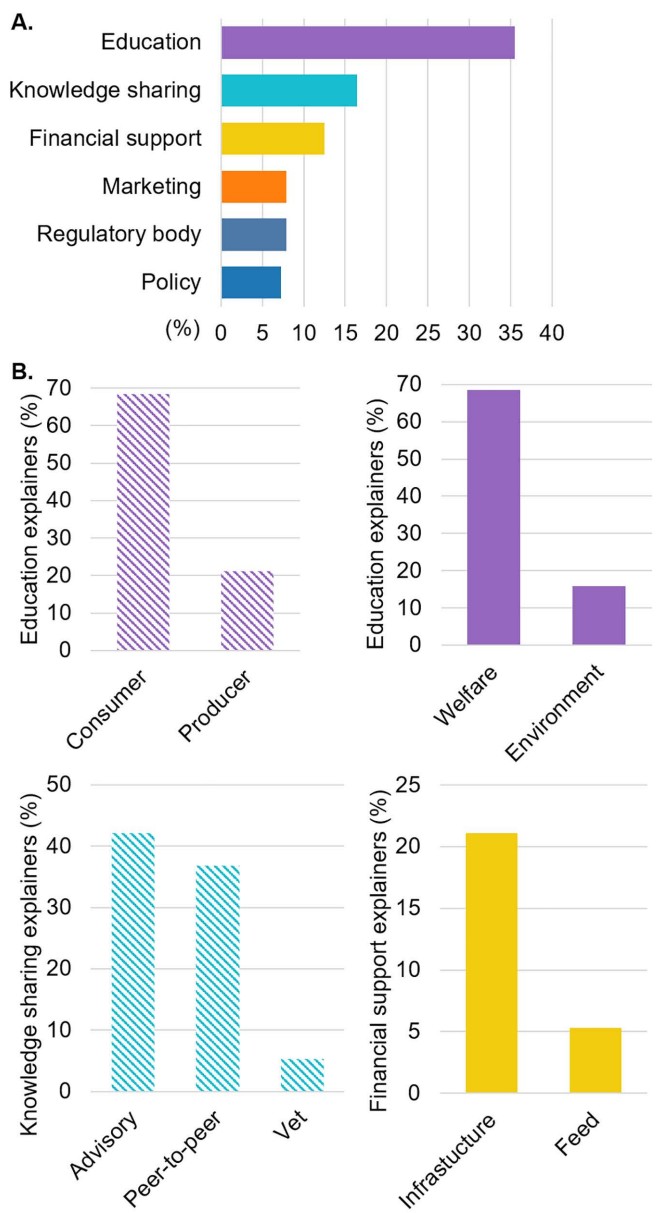

**Fig 6. Proportion of the 6 codes mostly used to describe the main changes needed to raise pigs outdoors (A) and their associated explainers (B.).** Dashed bars represent explainers related to stakeholders, and full bars represent explainers related to topics.

## Discussion

This study is the first of its kind on the island of Ireland and thus forms an invaluable source of information regarding the current state of play in terms of management of outdoor pigs, and the motivations and perceptions of the pig keepers involved in this sector. Due to the unknown number of outdoor pig farms in the island of Ireland, it was not possible to estimate the response rate level; nevertheless the quantity and quality of the information that was gathered provides a gateway towards further research and policy engagement with this community.

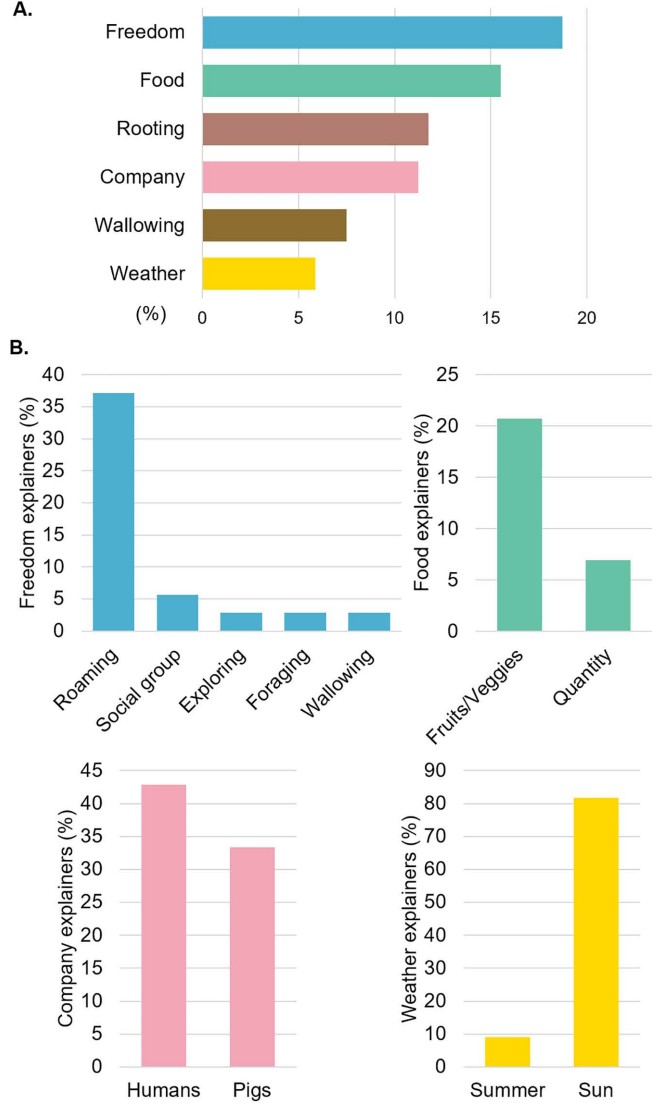

**Fig 7. Proportion of the 6 codes mostly used to describe the main things that respondents thought pigs enjoy in life (A) and their associated explainers (B).**

A significant finding was that outdoor managed pigs in Ireland are generally kept in farms that could be considered 'back-yard' in nature, given the low numbers of animals in each holding. This is in contrast to outdoor pig systems in several other European countries, where larger scale outdoor commercial operations are still a small proportion of the total pig sector, but more prevalent (e.g., 40% of UK sows are outdoor bred [21]; 23.53% extensive pig productions in Spain [22]). Thus, in-depth comparisons with these systems are not relevant in the context of outdoor Irish pig production. In fact, participants often described their farm as "small" or "smallholding". The nature of these systems is reflected in the fact that the most common reason selected for keeping pigs was for personal meat consumption, with just over half of respondents producing meat for sale (49/90), and of these, over 80% sold their meat directly from the farm. Likewise, over half of the farms had only one person responsible for looking after the pigs, and over a fifth sourced their pigs from advertisements. More than half of the participants bred pigs themselves, which is in fact also typical of larger commercial scale

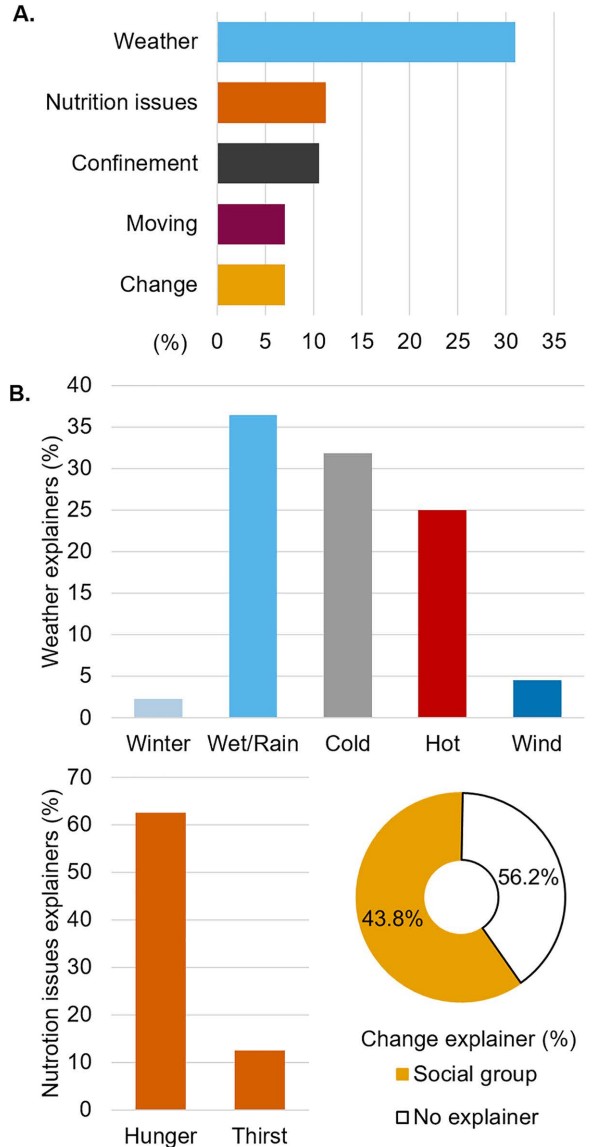

**A.**

**B.**

**Fig 8. Proportion of the 5 codes mostly used to describe the main things that respondents thought pigs least enjoy in life (A) and their associated explainers (B).**

enterprises. A significant finding was that the selection of slaughter facility was based upon proximity to the farm, and the facility's capability to slaughter small numbers of pigs; this highlights the importance of supporting the capability of the existing network of licenced slaughterhouses and/or victuallers to handle small numbers of pigs, and of a greater range of age and size than those typically produced commercially. Although "slaughter and processing" code was not reported as one of the main challenges (see S1 Text), in another study where 12 alternative pig farmers were interviewed in-depth, "processing the carcass" was described as a key challenge for outdoor pig farmers, and the use of mobile abattoirs to overcome the decrease in number of small abattoirs that lead to higher cost and poorer animal welfare (e.g., longer transport duration) was discussed [23].

Whether for personal consumption or for sale, the main reasons for keeping outdoor pigs were related to pigmeat production. It is possible that the reasons for this related to concerns about meat quality and traceability, as over 80% of participants considered that "safer, tastier and better quality meat" is "extremely important" when considering outdoor pig production. Traditional breeds, which were the most commonly kept, have higher intramuscular fat content, which contributes to the flavour and juiciness of the meat [24]. The three most commonly used breeds were Duroc, Oxford Sandy and Black, and Tamworth. Tamworth pigs have darker meat than Hampshire if reared on pasture, which is considered more desirable for the consumers [25]. Duroc pigs, while also typically used in commercial indoor farms, have a high fat content, and favourable sensory qualities [26]. These were commonly used to produce meat for sale, particularly with regard to male pigs. This could indicate the use of Duroc genetics to produce piglets from sows of traditional/rare breeds where meat quality or taste is not a main consideration [27], even though these breeds are well adapted to outdoor systems. Although Oxford Sandy and Black is highly used and anecdotally has "very high quality and flavour" meat [28], no literature describing the meat quality of this breed was found by the authors. Investigation of meat quality characteristics for this breed could be of use for the Irish outdoor pig industry, given its widespread use. "Food provenance" was also extremely important for the participants, in accordance with the significant demand from European pig meat consumers for traceability and transparency [29]. This could be a contributing factor towards the high proportion of participants selling directly from the farm, local butchers, farmers markets, or from their own online shop. These selling strategies fit with the European "Farm to Fork Strategy", and show the effort of the farmers to offer transparency to the consumers. This selling strategy can also be a consequence of most outdoor pig farmers being disinterested in selling meat at supermarkets due to a desire to have control of their supply chain, or being unable to due to the requirement for more meat than is possible to supply [23]. Moreover, participants that raised pigs for sale were all registered with the relevant Department of Agriculture, and more were raising pigs to an organic or GM free standard than participants that raised pigs for other purposes.

## Raising pigs outdoors for better animal welfare

There was little evidence of poor pig health in the survey, with less than 30% of respondents reporting evidence of any health condition or illness on their farm during the preceding 12 months. Incidence of respiratory symptoms reported by the respondents was low, which was also observed in other European organic pig farms [30], and which contrasts with indoor systems [13]. Contrarily, skin parasites were reported by nearly a third of participants, which is associated with outdoor pig farming [13] and can cause discomfort [31] and disease transmissions [32]. Similar to conventional systems, gastrointestinal parasites and lameness were among the most prevalent health disorders self-reported. Nevertheless, we subsequently found that gastrointestinal parasite prevalence was likely under-reported. While it was reported by only 3.6% of the respondents, a field study that we conducted on a subset of farms from this survey found that the most prevalent gastrointestinal parasites (*Eimeria*/*Isospora* spp. and strongyles) were detected on 80% of farms [33,34]. Participants seemed aware of measures that can be used to prevent these issues, reflected by the high proportion that performed "good hygiene and equipment maintenance", and "paddock rotation" (a strategy to reduce gastrointestinal parasites). Interestingly, less than a quarter of participants reported use of anthelmintics, antibiotics, vaccines, pain relief or antiseptics, which are the main preventative health strategies recommended and used in conventional pig farms in Ireland [35,36]. It is possible that this could be in part due to the fact that nearly half of the farms raised pigs to an organic or GM free standard. Interestingly, participants did not mention health management as a challenge (open question), despite 31% of participants selecting "unknown" when asked about their pigs' health conditions. The underestimation of health conditions or their lack of identification might be due to several factors, including the difficulty of assessing pigs raised in semi-natural environments [13], the inability of stock-owners to recognize or address welfare issues [37], and/or to the self-reported nature of this study methodology [38,39].

Participants identified both heat and cold stress as challenges to pig health under outdoor conditions, which ties in with some of the more commonly noted descriptors regarding what pigs least enjoy (e.g., wet/rain, cold, hot and windy

weather). Nearly all participants however stated they use "good animal care" as a preventative health measure. For example, they reported using "additional straw" on solid concrete or mucky areas, which increases insulation from wet and cold surfaces [13]. Participants also mostly raised traditional and/or rare breeds which are better adapted to outdoor conditions than "modern" genotypes [40,41].

Other factors that pigs do not enjoy fell under the category of nutrition issues (hunger and thirst). The widespread use of manually filled movable feeders suggests that animals may not have continuous ad-libitum access to feed. Pigs exposed to cold conditions have a higher requirement for feed [42], and thus it was positive to note that approximately two thirds of participants adapted feeding management practices to the season.

Regardless of whether pigs were kept for sale or for other purposes, animal welfare was considered "extremely important" to participants. This aligns with the results of Brajon et al. [14], who reported that promotion of multiple aspects of animal health and welfare was a significant driver for farmers to provide outdoor access for pigs. Consideration of the ability to express natural behaviours appeared to be a priority, as when asked what pigs enjoy, many of the code and explainers reflect behaviours or factors that are difficult to ensure indoors (e.g., "Freedom", "Rooting", "Wallowing", "Sun"). Contrarily, as stated above, the most identified code and its explainers capturing what pigs least enjoy were "weather" being "wet/rain" and/or "cold", indicating an awareness of the welfare risks associated with keeping pigs outdoors that are irrelevant in an intensive indoor system.

## Challenges and changes needed to promote outdoor pig farming in Ireland

Although the most common code identified to describe farms was "enjoyable", keeping pigs outdoors comes with challenges. As in Carroll et al. [23], feed cost was the most commonly reported challenge. In general, feed represents the main source of expenditure in pig farming systems [43], with organic ingredients and feed being even more expensive [44]. In our study, although some home-grown feed was used, feed merchants were the primary source of pig feed. For almost two fifths of producers, reliance on purchased organic feed likely explains why this challenge was mentioned. It is unsurprising that participants expressed the need for financial support not only for feed, but also for infrastructure. A lack of financial support was also reported by [14], where costs related to building expenses, debt load, and new regulations were identified as barriers to providing outdoor access to pigs. This aligns with our study, where nearly half of participants believed that keeping pigs outdoors is not at all, not so important, or neither important nor unimportant reason to reduce the feeding and buildings' maintenance costs.

Furthermore, participants emphasized the need for consumer education about animal welfare and the environment, as informed consumers can help farmers differentiate their products in the market [45]. As approximately a third of consumers valued a pig welfare label in Ireland [46], greater consumer awareness of farming systems could further support sustainable farming practices. Participants also reported a need for knowledge sharing from advisory services and/or peers. This need was also reported in a separate in-depth study where Irish pig farmers were interviewed individually. A lack of knowledge availability for alternative pig farming methods leads to farmers learning via trial and error approaches, and from others in the alternative pig farming community [23]. This community based approach is reflected in our study, as 29.4% and 9.4% of the participants stated that they sought advice from other outdoor farmers, and other sources that include societies when it comes to managing animal health, as well as half of the participants reporting being part of a society.

Finally, participants reported that they raise pigs outdoors because "it is more sustainable" (58.7%). Several produce home-grown feed, which reduces greenhouse gas emissions [47]. The contribution of pigs to the environment was also "extremely important" for the participants, as they can provide benefits to land management by controlling "bramble and invasive plants without ploughing and spraying" [23], and improve the biodiversity and soil health under paddock rotation management [48]. However, they also mentioned paddock management and soil quality (open question) as challenges related to the weather and land base. In studies done in France [14] and in the island of Ireland [23], farmers expressed

concerns about maintaining soil in good agricultural and environmental condition, emphasizing respectively that "outdoor farming is not feasible on any type of land", and how they adapt the animal and paddock managements around the seasons and weather conditions. In our study, 77% of participants reported having clay-based soil, which might increase the difficulty of maintaining good vegetation coverage, and risk of flooding, given that Ireland has one of the highest levels of rainfall in Europe [49]. However, even loamy and sandy soils, perceived by conventional Irish farmers to be more predominant in UK and more suitable for raising pigs outdoor [23], have been shown to have environmental challenges. A study conducted in England found that on farms with these soil types, the presence of pigs was responsible for bare soil, and high rainfall led to flooding which spread sediment, faeces, and pathogens [50]. In Mediterranean regions, although the rainfall is lower, soil erosion is also a challenge on sandy-loam farms with high animal stocking density [51].

## Conclusion

This study provides valuable insight into Ireland's outdoor pig farming sector, revealing its predominantly small-scale nature, with most pigs kept for personal consumption, and/or direct meat or pig sales. Farmers see animal welfare, meat quality, and food provenance as drivers in raising pigs outdoor and prioritized them, though challenges such as feed costs, financial support, and soil maintenance persist. Although the pigs' health in outdoor systems was reported as being generally good, health issues might be under-observed, suggesting a need for improved monitoring. Greater consumer education and financial support from the government could enhance the sector's sustainability. Future research should focus on breed-specific adaptation to outdoor conditions, meat quality and market demand, and environmental management strategies to support outdoor pig farming in Ireland, while ensuring economic viability and high welfare standards.

## Supporting information

**S1 Text. Overall questionnaire.** Questions annotated with * were the questions that had to be answered to include the participation in the study.
(PDF)

## Acknowledgments

We would like to thank Dr. Grace Carroll from Queen's University Belfast for advice in analysing the open questions of the survey.

## Author contributions

**Conceptualization:** Ophélie Menant, Laura Boyle, Siobhan Mullan, Fidelma Butler, Keelin O'Driscoll.

**Data curation:** Ophélie Menant.

**Formal analysis:** Ophélie Menant, Keelin O'Driscoll.

**Funding acquisition:** Laura Boyle, Keelin O'Driscoll.

**Investigation:** Ophélie Menant.

**Methodology:** Ophélie Menant, Laura Boyle, Siobhan Mullan, Fidelma Butler, Keelin O'Driscoll.

**Project administration:** Laura Boyle, Keelin O'Driscoll.

**Validation:** Laura Boyle, Keelin O'Driscoll.

**Writing – original draft:** Ophélie Menant.

**Writing – review & editing:** Ophélie Menant, Laura Boyle, Siobhan Mullan, Fidelma Butler, Keelin O'Driscoll.

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
