## [Decision Letter · Decision Letter 0]

23 Jul 2025

Dear Dr. Menant,

Thank you for submitting your manuscript to PLOS ONE. After careful consideration, we feel that it has merit but does not fully meet PLOS ONE’s publication criteria as it currently stands. Therefore, we invite you to submit a revised version of the manuscript that addresses the points raised during the review process.

We look forward to receiving your revised manuscript.

Kind regards,

Mehdi Borhani Zarandi

Academic Editor

PLOS ONE

Journal Requirements:

[This study was funded by the Department of Agriculture, Food and the Marine in Ireland (Grant number 2021R600).].

4. Thank you for stating the following in your manuscript:

[This study was funded by the Department of Agriculture, Food and the Marine in Ireland (Grant number 2021R600).]

[This study was funded by the Department of Agriculture, Food and the Marine in Ireland (Grant number 2021R600).]

Reviewers' comments:

Reviewer's Responses to Questions

**Comments to the Author**

1. Is the manuscript technically sound, and do the data support the conclusions?

Reviewer #1: No

Reviewer #2: Yes

Reviewer #3: Yes

2. Has the statistical analysis been performed appropriately and rigorously?

Reviewer #1: No

Reviewer #2: Yes

Reviewer #3: No

3. Have the authors made all data underlying the findings in their manuscript fully available?

Reviewer #1: No

Reviewer #2: Yes

Reviewer #3: No

4. Is the manuscript presented in an intelligible fashion and written in standard English?

Reviewer #1: Yes

Reviewer #2: Yes

Reviewer #3: Yes

Reviewer #1: This study is appropriate for internal and epidemiological journals. This study is appropriate for internal and epidemiological journals, as it offers valuable insights into the patterns and determinants of disease spread within specific populations. By leveraging both quantitative and qualitative data, it provides a comprehensive overview of health trends that can inform preventive strategies and intervention policies.

Reviewer #2: This paper provides a comprehensive review of outdoor pig farming systems in Ireland, a previously understudied area. Using an online questionnaire and analysis of qualitative and quantitative data, the authors were able to comprehensively identify farmers’ motivations, challenges, and needs in the sector. This study fills a gap in the literature on small-scale, decentralized pig farming systems and provides valuable insights for policymakers and industry practitioners.

I have some minor questions:

What factors have been identified as the main drivers for pig farming in open systems in Ireland and how do these relate to financial and infrastructural challenges?

Are there significant differences in pig health and welfare management between small and large farms in open systems in Ireland?

What is the role of traditional and rare pig breeds in the success of open systems in Ireland and how are these breeds adapting to climate conditions and market demands?

What solutions have been proposed to improve the economic and environmental sustainability of pig farming systems in Ireland?

Reviewer #3: "Pigs and Pasture: Drivers and characteristics of outdoor systems on the island of Ireland"

The study addresses a significant gap in the literature by characterizing outdoor pig production systems on the island of Ireland, a topic previously under-researched compared to intensive indoor systems. The focus on small-scale, outdoor systems provides novel insights into farmer motivations, management practices, and challenges, contributing to broader discussions on sustainable agriculture, animal welfare, and alternative farming models. The inclusion of both commercial and non-commercial producers adds depth to the analysis.

However, concerns and issues requiring revision should be addressed:

Sample Representativeness and Bias: The study relies on self-selected respondents (n=90), which may overrepresent farmers engaged in welfare-focused or organic production, skewing results. No response rate is calculable due to the unknown total population of outdoor pig farms, limiting generalizability.

Health data reliability: Self-reported health metrics (e.g., only 31% reporting health issues) conflict with field studies cited (e.g., 80% parasite prevalence in a subset of farms). This suggests underreporting or lack of farmer expertise in disease recognition. No objective health assessments (e.g., veterinary records, lab tests) were used, risking bias.

Methodological gaps: Open-ended question coding lacks inter-rater reliability metrics, raising questions about consistency in qualitative analysis. Co-occurrence analysis, while useful, does not account for confounding variables (e.g., farm size influencing challenges like feed costs).

Geographic and comparative limitations. Findings are specific to Ireland’s small-scale systems; comparisons to larger EU outdoor systems (e.g., UK, Spain) are minimal, reducing broader applicability. Climate/soil differences (e.g., Ireland’s high rainfall vs. Mediterranean systems) are noted but not deeply analyzed for their impact on challenges like paddock management.

Ambiguity in definitions: Organic/GM-free standard" is self-reported without verification of certification, potentially misclassifying farms. "Extreme weather" is not quantitatively defined (e.g., temperature thresholds, rainfall levels), making it subjective.

Policy and financial analysis superficiality: Calls for "financial support" and "consumer education" lack specificity (e.g., proposed subsidy structures, educational campaigns). No cost-benefit analysis of suggested interventions (e.g., mobile abattoirs) is provided.

Data availability timing

Data will be available post-publication, hindering immediate reproducibility. A preprint or supplemental upload at submission would strengthen transparency.

Recommended Revisions: Thus, address health data discrepancies by cross-referencing self-reports with veterinary records or farm visits. Clarify qualitative coding methodology (e.g., inter-rater agreement). Expand comparative discussion to EU outdoor systems. Define key terms (e.g., "extreme weather") operationally. Propose concrete policy measures (e.g., grant schemes, training programs).

These revisions would enhance rigor and applicability without altering the study’s core contributions.

**Do you want your identity to be public for this peer review?** For information about this choice, including consent withdrawal, please see our Privacy Policy

Reviewer #1: No

Reviewer #2: No

Reviewer #3: No

---

## [Author Response · Author response to Decision Letter 1]

11 Sep 2025

Journal Requirements:

The new version of the manuscript follows the PLOS ONE’s style.

Details regarding consent are included in lines 88-90 in the materials and methods section. The participants were asked to confirm consent by ticking a box at the beginning of the online survey. We have included the information that participants could not progress to the survey unless this box was ticked.

[This study was funded by the Department of Agriculture, Food and the Marine in Ireland (Grant number 2021R600).].

The statement regarding the financial disclosure is in the cover letter.

4. Thank you for stating the following in your manuscript:

[This study was funded by the Department of Agriculture, Food and the Marine in Ireland (Grant number 2021R600).]

[This study was funded by the Department of Agriculture, Food and the Marine in Ireland (Grant number 2021R600).]

The funding statement should appear as followed: “This study was funded by the Department of Agriculture, Food and the Marine in Ireland (Grant number 2021R600). The funders had no role in study design, data collection and analysis, decision to publish, or preparation of the manuscript.” The cover letter contains the request for this amendment.

The data are now available at https://opendata.teagasc.ie/dataset/survey_production_health_welfare_outdoor_pig_farms_ireland.

Comments to the Author

Reviewer #1: This study is appropriate for internal and epidemiological journals. This study is appropriate for internal and epidemiological journals, as it offers valuable insights into the patterns and determinants of disease spread within specific populations. By leveraging both quantitative and qualitative data, it provides a comprehensive overview of health trends that can inform preventive strategies and intervention policies.

We thank reviewer #1 for highlighting the valuable information about pig health raised on outdoor systems that our work has generated.

Reviewer #2: This paper provides a comprehensive review of outdoor pig farming systems in Ireland, a previously understudied area. Using an online questionnaire and analysis of qualitative and quantitative data, the authors were able to comprehensively identify farmers’ motivations, challenges, and needs in the sector. This study fills a gap in the literature on small-scale, decentralized pig farming systems and provides valuable insights for policymakers and industry practitioners.

We would like to thank the reviewer #2 for the thoughtful and constructive feedback on our work. We hope that our changes in the manuscript answer the reviewer’s questions.

I have some minor questions:

What factors have been identified as the main drivers for pig farming in open systems in Ireland and how do these relate to financial and infrastructural challenges?

The survey identified several reasons for pig keeping in outdoor pig farms, as well as some of the drivers as to why respondents keep pigs this way. The purpose for which respondents keeping pigs in outdoor farms is described in L173-178, and Figure 1. The reasons why they feel keeping pigs this way is important is described in the section from L323 – 327. These include animal welfare, meat quality, traceability and environmental sustainability.

We did not carry out an analysis to investigate whether there is a relationship between what respondents reported as being their main drivers for keeping pigs outdoors, and the answers to the questions about challenges and changes needed at infrastructural level. The main challenges identified are listed from L339 – 348, and Figure 5A, and the changes from L355-364. However, education was the change that is needed that was most commonly included, in relation to consumers, farmers, and regarding animal welfare and the environment. Challenges mainly related to finances. Thus, it is possible that respondents consider that improving education amongst consumers regarding the drivers for their production systems could help to support their businesses by resulting in more people purchasing their meat (L523-525). In the conclusion (Lines 558-560) we have also included the following statement: “Farmers see animal welfare, meat quality, and food provenance as drivers in raising pig outdoor and prioritized them, though challenges such as feed costs, financial support, and soil maintenance persist.”

Are there significant differences in pig health and welfare management between small and large farms in open systems in Ireland?

As described in the study, outdoor pig systems in Ireland are small scale (median 21 animals). We were unable to identify any large scale outdoor farming enterprises on the Island, and consequently no farm scale comparison on animal health could be performed. Authors could not find any study performed in other countries that compare small and large outdoor farms to rate the pig health reported in their study and former ones.

What is the role of traditional and rare pig breeds in the success of open systems in Ireland and how are these breeds adapting to climate conditions and market demands?

The result of this study does not allow making a direct link between the choice of raising traditional pig breeds and the success/sustainability of outdoor systems in Ireland. This interesting question should be further studied, so our conclusion has been modified to “Future research should focus on breed-specific adaptation to outdoor conditions, meat quality and market demand, and environmental management strategies to support outdoor pig farming in Ireland, while ensuring economic viability and high welfare standards.” (Line 563-566)

What solutions have been proposed to improve the economic and environmental sustainability of pig farming systems in Ireland?

This study comprised the first time ever that there has been a nationally funded research project looking at small/outdoor pig production – to date there has not even been a national advisory or certification programme for this sector, to support farmers. In fact, this is one of the reasons that the project was funded by the Department of Food, Agriculture, and the Marine (DAFM), to obtain information about how this sector is structured, and the challenges that it faces, to help identify strategies to support it. As part of the wider project we have included a task to develop an industry led roadmap towards higher welfare pig production on the island of Ireland. This is led by Prof. Siobhan Mullan, who is one of the authors of this paper. The group includes stakeholders from across the entire pig sector, including outdoor farmers, and representatives from DAFM. The aim is to identify strategies as you suggest, to help improve the sustainability of the sector, while hopefully improving welfare standards, and set aims and targets that could be met in 5, 10, 20 years etc. The project is still ongoing, but the final report and roadmap that we produce from this and other tasks will comprise the first step in suggesting solutions nationally. The Irish Animal Welfare Strategy (2021-2025) incorporates the guiding principles of working in partnership, science-based policymaking, improving education, consistent evaluation, and an effective regulatory system, and as such the project feeds into this in a structured fashion. Our ambition is that our outputs will help to guide policies to support continuation and expansion of higher welfare, and outdoor pig production in Ireland.

Reviewer #3: "Pigs and Pasture: Drivers and characteristics of outdoor systems on the island of Ireland"

The study addresses a significant gap in the literature by characterizing outdoor pig production systems on the island of Ireland, a topic previously under-researched compared to intensive indoor systems. The focus on small-scale, outdoor systems provides novel insights into farmer motivations, management practices, and challenges, contributing to broader discussions on sustainable agriculture, animal welfare, and alternative farming models. The inclusion of both commercial and non-commercial producers adds depth to the analysis.

We would like to thank the review #3 highlighting the relevance of our study and the variety of topics addressed in it. We hope that the changes made in the manuscript clarified the reviewer’s concerns about the data collection and analyses.

However, concerns and issues requiring revision should be addressed:

Sample Representativeness and Bias: The study relies on self-selected respondents (n=90), which may overrepresent farmers engaged in welfare-focused or organic production, skewing results. No response rate is calculable due to the unknown total population of outdoor pig farms, limiting generalizability.

As the reviewer indicated, our study is based on voluntary participations that might bias results; however, participants were not selected based on their engagement on animal welfare nor organic production. As the study focused on outdoor pig farm systems the selection criteria was that respondents had to raise pigs outdoors as stipulated lines 127-130. Reasons for not including responses in the analyses were explained lines 130-131. We have acknowledged that it is not possible to estimate response rate at the beginning of the discussion, L404 - 406.

Health data reliability: Self-reported health metrics (e.g., only 31% reporting health issues) conflict with field studies cited (e.g., 80% parasite prevalence in a subset of farms). This suggests underreporting or lack of farmer expertise in disease recognition. No objective health assessments (e.g., veterinary records, lab tests) were used, risking bias.

Recommended Revisions: Thus, address health data discrepancies by cross-referencing self-reports with veterinary records or farm visits.

The reviewer highlighted a common bias of survey studies, which is the subjectivity, and/or awareness of the participants when answering. We have changed the wording in L461 to reflect that the lack of evidence many health issues was based only upon survey results. This bias is also discussed from lines 467-472. In brief, following on from this survey, we carried out a study whereby we visited a sub-set of farms to assess the pigs’ health and welfare. Ultimately this will enable us to compare the differences in results between the self-reported data in the survey, and the field work. To date, we have published the first paper from this work, on the subject of gastrointestinal parasites (Senanayake et al., 2025), and found that gastrointestinal problems may be under-reported. More data collected during the field work (e.g. body condition, lameness, lesion scores etc.) has been analysed, and a manuscript is currently in preparation. These animal based measures will be compared and discussed in the context of the survey results in that paper.

Reference:

Senanayake, S.N., Boyle, L., O’Driscoll, K., Menant, O. and Butler, F. 2025. Effects of season, age and parasite management practices on gastro – intestinal parasites in pigs kept of outdoors pig farms in Ireland. Irish Veterinary Journal. 78:12 https://doi.org/10.1186/s13620-025-00297-0

Methodological gaps: Open-ended question coding lacks inter-rater reliability metrics, raising questions about consistency in qualitative analysis. Co-occurrence analysis, while useful, does not account for confounding variables (e.g., farm size influencing challenges like feed costs).

Recommended Revisions: Clarify qualitative coding methodology (e.g., inter-rater agreement).

More details about the answers format of the open questions analysed, and the coding methodology were added in the Materials and methods section of the manuscript in L108-109, and L139-148.

In the analysis section the sentence “Due to the diversity in farm systems, no farm characteristics were included as confounding variables in the co-occurrence analyses.” was added to explicitly inform that no confounding variables were used for the co-occurrence analysis (L161 – 163).

Geographic and comparative limitations. Findings are specific to Ireland’s small-scale systems; comparisons to larger EU outdoor systems (e.g., UK, Spain) are minimal, reducing broader applicability. Climate/soil differences (e.g., Ireland’s high rainfall vs. Mediterranean systems) are noted but not deeply analyzed for their impact on challenges like paddock management.

Recommended Revisions: Expand comparative discussion to EU outdoor systems.

As highlighted by the reviewer, the wide diversity of outdoor farms systems and climate conditions across Europe make comparison between systems challenging. We did not identify any large-scale outdoor pig farming operations that are comparable to the systems that are commonly found in for example the UK, or Spain. For this reason, we were unable to include a comparative discussion between the systems in our findings and those observed in Europe. We have noted this in L414-415. (Thus in-depth comparisons with these systems are not relevant in the context of outdoor Irish pig production).

We also added

---

## [Decision Letter · Decision Letter 1]

7 Jan 2026

Pigs and pasture: Drivers and characteristics of outdoor systems on the island of Ireland

PONE-D-25-23811R1

Dear Dr. Menant,

We’re pleased to inform you that your manuscript has been judged scientifically suitable for publication and will be formally accepted for publication once it meets all outstanding technical requirements.

Kind regards,

Grzegorz Woźniakowski, Full professor, PhD, ScD

Academic Editor

PLOS One

Additional Editor Comments (optional):

Reviewers' comments:

Reviewer's Responses to Questions

**Comments to the Author**

Reviewer #1: (No Response)

Reviewer #3: All comments have been addressed

Reviewer #4: All comments have been addressed

2. Is the manuscript technically sound, and do the data support the conclusions?

Reviewer #1: Partly

Reviewer #3: Yes

Reviewer #4: Yes

3. Has the statistical analysis been performed appropriately and rigorously?

Reviewer #1: Yes

Reviewer #3: N/A

Reviewer #4: Yes

4. Have the authors made all data underlying the findings in their manuscript fully available?

Reviewer #1: Yes

Reviewer #3: Yes

Reviewer #4: (No Response)

5. Is the manuscript presented in an intelligible fashion and written in standard English?

Reviewer #1: Yes

Reviewer #3: Yes

Reviewer #4: (No Response)

Reviewer #1: Dear Authours

This manuscript covers a timely and relevant topic by characterizing outdoor pig farming systems in Ireland, a sector that has been under-researched compared to conventional indoor production. The study’s strengths include a clear research gap, an appropriate survey design, and the combination of both quantitative and qualitative data. The findings provide useful insights regarding farmers’ motivations, challenges, and barriers in extensive pig production. Importantly, the focus on animal welfare and sustainable practices is a major strength.

However, several limitations deserve attention. First, the self-selected sampling and lack of response rate calculation reduce the generalizability of findings. Second, reliance on self-reported health and production data raises concerns regarding accuracy, particularly as field studies suggest higher disease prevalence than respondents reported. The absence of objective measures such as veterinary records or health checks further weakens the conclusions regarding animal health. Additionally, some key survey terms (like “inclement weather” and “organic standard”) would benefit from operational definitions. Finally, while the discussion recognizes the need for policy and financial support, recommendations remain broad; more concrete suggestions would increase the manuscript’s impact for stakeholders.

Overall, this work addresses an important knowledge gap and offers a valuable platform for future research and policy development, but would benefit from enhanced methodological detail, critical discussion of sample bias, and greater specificity in recommendations.

Best regard

Reviewer #3: I have reviewed the authors’ revised manuscript and their responses to the second-round comments. The authors have adequately addressed the key concerns raised, particularly regarding sample representativeness, self-reported health data, methodological transparency, and the scope of comparative discussion.Concerns regarding the reliability of self-reported animal health data have been addressed by clarifying that findings are based solely on survey responses, explicitly discussing potential underreporting, and referencing follow-up fieldwork conducted by the authors.

Limitations inherent to survey-based research are now clearly acknowledged, and the revisions improve clarity and transparency without overstating the findings.

Reviewer #4: (No Response)

**Do you want your identity to be public for this peer review?** For information about this choice, including consent withdrawal, please see our Privacy Policy

Reviewer #1: No

Reviewer #3: No

Reviewer #4: No

---

## [Editor Report · Acceptance letter]

PONE-D-25-23811R1

PLOS One

Dear Dr. Menant,

I'm pleased to inform you that your manuscript has been deemed suitable for publication in PLOS One. Congratulations! Your manuscript is now being handed over to our production team.

Kind regards,

on behalf of

Prof. Grzegorz Woźniakowski

Academic Editor

PLOS One